# Bioresorbable Magnesium-Based Stent: Real-World Clinical Experience and Feasibility of Follow-Up by Coronary Computed Tomography: A New Window to Look at New Scaffolds

**DOI:** 10.3390/biomedicines11041150

**Published:** 2023-04-11

**Authors:** Chadi Ghafari, Nicolas Brassart, Philippe Delmotte, Philippe Brunner, Sarah Dghoughi, Stéphane Carlier

**Affiliations:** 1Department of Cardiology, Research Institute for Health Sciences and Technology, University of Mons (UMONS), 7000 Mons, Belgium; 2CHU Ambroise Paré, 7000 Mons, Belgium

**Keywords:** coronary computed tomography angiography (CCTA), bioresorbable scaffold, DREAMS-2G

## Abstract

(1) Background: The diagnostic accuracy of coronary computed tomography angiography (CCTA) for coronary artery disease (CAD) has greatly improved so CCTA represents a transition in the care of patients suffering from CAD. Magnesium-based bioresorbable stents (Mg-BRS) secure acute percutaneous coronary intervention (PCI) results without leaving, in the long term, a metallic caging effect. The purpose of this real-world study was to assess clinical and CCTA medium- and long-term follow-up of all our patients with implanted Mg-BRS. (2) Methods: The patency of 52 Mg-BRS implanted in 44 patients with de novo lesions (24 of which had acute coronary syndrome (ACS)) was evaluated by CCTA and compared to quantitative coronary angiography (QCA) post-implantation. (3) Results: ten events including four deaths occurred during a median follow-up of 48 months. CCTA was interpretable and in-stent measurements were successful at follow-up without being hindered by the stent strut’s “blooming effect”. Minimal in-stent diameters on CCTA were found to be 1.03 ± 0.60 mm smaller than the expected diameter after post-dilation on implantation (*p* < 0.05), a difference not found in comparing CCTA and QCA. (4) Conclusions: CCTA follow-up of implanted Mg-BRS is fully interpretable and we confirm the long-term Mg-BRS safety profile.

## 1. Introduction

Continuous techniques and scaffold advances have come a long way in coronary artery disease (CAD) treatment since Andreas Gruntzig’s pioneering work some 40 years ago [1,2]. The introduction of the first bare metallic stents (BMS) in the 1980s revolutionized percutaneous coronary interventions (PCI). Currently, intracoronary stents remain the gold standard for the treatment of coronary artery lesions [3]. Despite offering major advances regarding stent material and design along with thinner struts, lower neointimal hyperplasia, restenosis rates and major adverse cardiac events (MACE), new generation drug-eluting stents (DES) still present limitations [4,5]. They leave a foreign metallic body within the vessel, presenting the potential for an adverse host reaction to a foreign body such as neointimal hyperplasia leading to in-stent restenosis or thrombosis, limited late lumen enlargement and lack of vasomotion [6]. It also presents physical restraints to the vessel physiology along with a “blooming effect” defined as stent strut beam hardening artifacts, resulting in artificial luminal narrowing and decreased intraluminal attenuation during non-invasive imaging with coronary computed tomography angiography (CCTA).

The appealing concept of a bioresorbable scaffold (BRS) aims to provide performance equivalent to existing DES by the delivery of a transient coronary device that would secure the acute results of percutaneous coronary intervention (PCI), preventing acute recoil and constrictive remodeling. Furthermore, the ideal BRS would elute an antiproliferative agent, avoiding neointimal hyperplasia before the biodegradation of its scaffold [7,8]. A more refined lesion preparation and implantation technique has been advocated due to the lower radial force, the higher crossing profile and the limited expansion diameter of a BRS [9]. It was also suggested that a BRS would be a good alternative to DES for the treatment of acute coronary syndromes (ACS) due to the underlying pathophysiologic characteristics of the disease [7,10]. The vulnerable soft nature of the plaque without extensive calcifications, along with the presence of thrombus and the younger age of the patients, represent a favorable terrain for a BRS implantation. Polymeric BRS implantation has been previously evaluated in ACS with varied results [11,12,13]. Moreover, BRS also present potential benefits for noninvasive X-ray imaging evaluation, such as CCTA scan, since the struts of the stent are not in “pure” metal and do not produce any “blooming” effect [6].

However, the daily clinical use of BRS nowadays remains limited secondary to safety concerns [14]. The first-generation BRS stents made of poly-L-lactic acid demonstrated higher rates of device failure, thrombosis and target lesion revascularization (TLR) as compared to metallic DES [1,6,15]. Nevertheless, metallic bioresorbable scaffolds are still under improvement and investigation (Table 1). DREAMS-2G is a second-generation magnesium-based BRS (Mg-BRS) that was developed as a transition in the care of patients suffering from CAD and it received the CE mark in 2016. It was designed using a slowly absorbable magnesium alloy backbone arranged in a geometrical 6-crown 2-link design to achieve sufficient radial force and uniform vessel coverage. The entire resorption process is speculated to last around 12 months [16,17]. The backbone is coated with 7 µm of a biodegradable poly-L-lactic-co-glycolic acid (PLGA) polymer BIOlute loaded with a calibrated release dose of sirolimus. DREAMS-2G showed promising results in the BIOSOLVE II-IV registries [18,19,20]. More recently, the MAGSTEMI evaluated the performance of the Mg-BRS versus the sirolimus-eluting DES in the context of ST-elevation myocardial infarction (STEMI). A significantly higher rate of late luminal loss and restenosis was found in the Mg-BRS arm [21].

On the other hand, advances in coronary computed tomography angiography (CCTA) imaging have facilitated a more accurate interpretation of noninvasive coronary imaging [22]. CCTA was shown to have a high quantitative and qualitative diagnostic accuracy as compared to quantitative coronary angiography (QCA) [23]. In patients with suspected anginal symptoms, CCTA was shown to help establish the diagnosis, aid in targeting interventions and reduce future risk of myocardial infarction [24]. It also permits the accurate evaluation of previously implanted stents suspected of in-stent restenosis [25,26]. As such, CCTA is drastically changing the way we evaluate CAD patients, offering new diagnostic and follow-up opportunities.

We sought to report short and long-term clinical and noninvasive imaging results of CAD patients treated at our institution with at least one Mg-BRS who were eventually scheduled to undergo a CCTA at any time post-implantation.

## 2. Materials and Methods

### 2.1. Study Design

This is a retrospective, monocentric, observational cross-sectional case series evaluating patients treated by at least one Mg-BRS. The Institutional Review Board (IRB) at the Centre Hospitalier Universitaire et Psychiatrique de Mons-Borinage (CHUPMB)—Belgium reviewed and approved the study protocol on 13 January 2022 (reference number HAP-2021-034). The board waived the requirement for written consent. Major adverse cardiovascular events (MACE) were defined as cardiovascular death, target vessel myocardial infarction (TV-MI), target vessel revascularization (TV-R) and clinically driven target lesion revascularization (CD-TLR). We retrospectively collected and assessed treated patients’ demographics and clinical information, including laboratory and procedural characteristics along with, when available, follow-up visits.

The primary endpoint was to determine the diagnostic accuracy of CCTA for assessing in-scaffold luminal dimensions as compared to quantitative coronary angiography at the time of the procedure, as well as the feasibility to detect in-stent restenosis. The secondary endpoint was to report long-term clinical follow-up of all our patients treated by at least one DREAMS-2G scaffold.

### 2.2. Quantitative Coronary Angiography Analysis

QCA analysis was performed at an independent core laboratory (UMONS CoeurLab) using a validated QCA software (CAAS v. 8.4 system, Pie Medical, Maastricht, The Netherlands). The operator was blinded to CCTA results. Proximal and distal vessel diameter and surface area were determined as well as lesion and in-stent minimal surface area (MSA) and diameter (MLD). In-stent mean proximal to distal surface area was calculated as the average of the proximal and distal surface area at the stent edge ((Proximal + Distal Surface Area)/2).

### 2.3. Coronary Computed Tomography Angiography Acquisition and Analysis

Assessment of stent patency was assessed by 240-slice Toshiba CCTA scans (Canon Medical Systems Corporations, Otawara, Tochigi, Japan) using standard acquisition techniques, including 6 hours fasting, heart rate control using beta-blockers when indicated and tube settings depending on the patient’s body mass index. Images were reconstructed and analyzed by two independent physicians at 0.5 mm increments. After cross-sectional vessel reconstruction, implanted stents were identified using the two well-radiopaque stent markers. Automatic segmentation of the vessel lumen was performed for the measurements and manual correction was made when needed. The maximal and minimal lumen area (MLA), proximal and distal surface areas and diameters as well as the in-stent minimal and maximal diameters were determined for each scaffold. Mean proximal to distal surface area was calculated.

### 2.4. Statistical Analysis

Categorical variables were described as frequency rates and percentages, and continuous variables were described as mean ± SD. Proportions for categorical variables were compared using Fisher’s exact test, χ^2^ and one-way ANOVA. The Mann–Whitney rank sum test, Kruskal–Wallis test and an independent t-test were used to compare continuous variables. A Wilcoxon signed-rank test was used to compare the median of differences. Agreement between measurement modalities was assessed by the Passing–Bablok and Bland–Altman methods. Long-term survival was represented by a Kaplan–Meier analysis. Statistical analyses were performed using SPSS software version 23.0 (IBM, Armonk, NY, USA) and MedCalc software version 20.125 (Ostend, Belgium).

## 3. Results

### 3.1. Study Population

Between January 2017 and July 2020, a total of 52 Mg-BRS were implanted to treat 50 de novo coronary lesions in 44 patients who underwent primary PCI (Table 2). The mean patient age was 56.3 ± 7.7 years and there were 36 males (81.8%). Twenty-four (54.5%) patients presented with ACS, among which 9 patients (20.9%) presented with ST-elevation myocardial infarction (STEMI). Intravascular ultrasound (IVUS) was used in 9 patients (20.5%). ACS Patients presenting with ACS tended to be younger (*p* < 0.05), males (*p* = 0.05) and smokers (*p* < 0.05). A majority of left anterior descending (LAD) arteries (63.5%) were treated. Mild calcifications were found in 19 lesions (37.3%) overall. Lesion classification, according to the ACC/AHA classification, was type A in 22 of the cases (42.3%). Pre-dilation was performed in all but three patients. The mean implanted stent diameter size was 3.15 ± 0.23 mm, inflated at a mean pressure of 14.9 ± 1.36 atm. Post-dilation was performed using a non-compliant balloon in all but three STEMI lesions (93.8%) at the discretion of the operator. The mean post-dilation balloon diameter was 3.4 ± 0.3 mm, inflated at a mean pressure of 18.5 ± 2.4 atm. This reflected the adherence of the operators to the predefined Mg-BVS implantation technique recommendations [27,28].

### 3.2. Clinical and Radiological Outcome

Clinical outcomes and follow-up were obtained from the patient records of all patients at 30 days, then at 6, 12, 24 and 48 months. The longest follow-up available to date is 70 months, with a median of 47.9 [28.9–56.0] months. Cardiac death, TV-MI, TV-R and CD-TLR rates were comparable among both ACS and elective groups: ten events (22.7%) were noted (four in the elective group and six in the ACS group). Three reported deaths (12.5%) were noted in the ACS group on days 3, 97 and 895 post-implantation. These were attributed to cardiac tamponade and two deaths of unknown origin, respectively (possible stent thrombosis following the ARC definition). In addition, one death was reported in the elective group attributed to COVID-19 infection complications 530 days post-implantation. Survival curves are shown in Figure 1. Definite stent thrombosis by angiography occurred in one patient (4.2%) on day 37 post-implantation. This patient, who presented initially with an ACS, was still on dual antiplatelet therapy. Three CD-TLRs were reported in three patients (12.5%) of the ACS arm at 2, 8 and 52 months, respectively. IVUS images showed in-stent neointimal hyperplasia treated by one DES implantation and severe plaque burden under the struts of the stent (Figure 2 and Figure 3). One lesion was treated surgically, whereas the others showed angiographic as well as functional in-stent restenosis and were treated by DES implantation. Overall event rates were not statistically significant between both arms (*p* = 0.734). Patients were on a dual antiplatelet regimen in both arms with a significantly higher tendency for ticagrelor in the ACS arm (*p* = 0.003). A detailed analysis of events in the ACS arm failed to show a statistically significant difference between the different sub-groups, as given in Table 3.

Results of the quantitative coronary analysis (QCA) of the index procedures are given in Table 4. There was no statistically significant difference found on QCA between the elective and ACS groups; however, a statistically significant difference was found between minimal vessel surface area post-stenting and the mean proximal to distal stent area (*p* < 0.05).

CCTA was performed in 36 out of the 44 patients (81.8%), out of which, 32 (88.9%) were interpretable and measured (Figure 4). Calcifications and/or artifacts prevented the analysis of the remaining studies. Stent resorption was noted in all patients and stents were identified using the two stent tantarums. The mean time between the index procedure and CCTA was 6.4 ± 3.7 months. The minimal in-stent surface area was 6.5 ± 2.0 mm^2^ and the mean proximal to distal in-stent surface area was 8.55 ± 2.3 mm^2^ (Table 5). An in-stent restenosis was found on a follow-up CCTA with an MSA intrastent of 1.1 mm^2^ and a reference vessel lumen area of 5.5 mm^2^. The Mg-BRS was implanted in an ACS patient who remained asymptomatic with no signs of ischemia on dobutamine stress echocardiography. A significant statistical difference was found between MSA and the mean proximal to distal in-stent surface area (*p* < 0.05). Minimal in-stent diameters on CCTA were found to be 1.03 ± 0.60 mm smaller than the expected stent diameter after post-dilation after implantation (*p* < 0.05) using the post-dilation balloon diameters as reference.

In 32 scaffolds, the mean difference between QCA and CCTA in MLD and MSA were −0.08 (limits of agreement (LOA) −1.29 to 1.46) and −2.84 (LOA −2.3 to 8.0) mm, respectively. The mean minimal in-stent diameter by CCTA was 2.38 ± 0.56 mm while the post-PCI QCA minimal lumen diameter was 2.4 ± 0.5 mm, a non-significant statistical difference. Overall, there was no agreement between QCA and CCTA on a Passing–Bablok regression analysis (Figure 5).

## 4. Discussion

The appealing concept of BRS has received a lot of enthusiasm in the last two decades, with the advantage of a scaffold providing radial strength in the acute phase and dissolving after a period of time, hence reducing the long-term complications of DES [16,29]. It is important to keep in mind that the first BRS were implanted more than 20 years ago by Tamai and Igaki [30]. Different concepts of bioresorbable scaffolds with respect to material, design and drug elution have been studied [31]. Although early-generation BRS results have been disappointing, novel-generation BRS including Mg-BRS showed promising results as compared to DES [32,33]. Our results report one of the longest follow-ups in a real-world setting of a cohort of unselected elective and ACS patients and support this new paradigm in percutaneous coronary interventions. As illustrated, the possibility to non-invasively follow up these stents by CCTA illustrates an interesting transition in cardiovascular care.

The main findings of this study are: first, Mg-BRS did not hinder in-stent lumen visualization measurements on CCTA. Second, Mg-BRS exhibited a trend for a higher rate of MACE in the setting of ACS although it did not reach a significant difference in our limited number of patients. Third, device-related thrombotic events were low in our cohort. Finally, intraluminal measurements using QCA and CCTA differed significantly, with a significant loss of in-stent diameter on CCTA.

CCTA has gained much interest in the past years as a noninvasive method for the evaluation of patients with CAD and was adopted by the guideline-directed workup of patients suspected of having CAD [34]. Advances in CCTA technology allow better luminal evaluation as well as plaque characterization, with contemporary studies showing high sensitivity and specificity to exclude obstructive lesions [35,36,37]. Moreover, the polymeric scaffold of BRS does not produce beam-hardening artifacts, hence minimizing the blooming effect and allowing CCTA to have higher diagnostic accuracy to detect in-scaffold lumen obstruction [38,39,40].

To the best of our knowledge, this is the largest cohort of patients implanted with a Mg-BRS evaluated by CCTA. The diagnostic accuracy of CCTA for the detection of bioresorbable scaffold obstruction and luminal dimensions has already been established using older-generation scaffolds [38]. Previously, the proof of concept was reported in one patient with a follow-up of 7 months, and case series of two patients by Wong et al. and eight patients by Salinas et al. who reported good quantitative luminal dimensions, as well as plaque composition analysis at a median CCTA follow-up of 12.7 months [41,42]. In our cohort, CCTAs were fully interpretable in a high proportion of patients at a shorter follow-up interval, with the absence of any blooming effect secondary to stent struts allowing for precise assessment and follow-up of implanted Mg-BRS, as well as precise quantitative and qualitative assessment. In-stent minimal diameter by follow-up CCTA did not differ from post-PCI angiographic minimal lumen diameter, showing a low in-scaffold late loss in line with the reported results at 12 months in BIOSOLVE II [18].

Our data are also in line with the MAGSTEMI, PRAGUE-19 and BEST-MAG studies concerning the safety of use of BRS in the setting of ACS all with a slightly higher rate of MACE [21,43,44]. Although our data failed to show any statistically significant difference in the rate of MACE between the ACS and elective groups, nevertheless, a higher number of events was reported in the ACS group. This could be attributed to the small sample size present in our cohort. A definite stent thrombosis was reported in our cohort; however, stent thrombosis has been rarely reported in a Mg-BRS scaffold with one (1.4%) definite thrombosis reported in the MAGSTEMI trial and five (0.5%) in the BIOSOLVE-IV registry [21,45]. Findings of these limited sample size reports could be secondary to more complex lesions as compared to earlier studies. On the other hand, a small registry of patients presenting with STEMI and treated with an Mg-BRS, as well as a pooled analysis of BIOSOLVE II and III at 3 years, failed to report any Mg-BRS thrombosis [46,47].

We report a higher number of MACE than the recent 8.0% TLF published at 5 years follow-up in the BIOSOLVE II study [33]. This is a reminder of the differences in patients included in studies and real-world experience. Caution remains important before applying new therapies tested in controlled studies for all patients.

In-scaffold luminal dimension evaluation has already been compared by QCA and CCTA, showing similar diagnostic accuracy [38]. Our QCA and CCTA measurements were performed, on average, 6 months apart and this might explain the lack of agreement reported. However, we confirm the feasibility of measuring BRS luminal areas by CCTA and demonstrate accurate estimation of a loss of in-stent minimal diameter.

Whereas dual antiplatelet therapy (DAPT) is recommended for at least 6 and 12 months post DES or BMS implantation in the elective and ACS setting, DAPT post-BRS implantation remains currently recommended ≥ 12 months independent of implantation setting due to a higher rate of late stent thrombosis [48]. With an increasing number of reports lately towards shorter DAPT duration, data concerning BRS remain scarce and inconclusive, and a possible extension of treatment beyond 12 months could be reasonable, extending from larger DES trials [49].

Our study highlights the safety of BRS implantation in a real-world simple, soft and mildly calcified de novo lesion, as recommended in BRS studies, allowing for a transient radial force securing acute PCI results and resorbing with time. Furthermore, our data underline the feasibility of CCTA follow-up of BRS, a non-invasive imaging tool, hence reducing the need for invasive coronary angiography for in-stent restenosis detection.

### Limitations

This was a retrospective study on a relatively small cohort. The time frame of CCTA follow-up was short for assessing long-term outcomes. The observational nature of our study conveys its limitations. CCTA quality remains a paramount point to consider. Factors such as high heart rate, arrhythmias, calcifications and high body index will negatively impact CCTA interpretation.

## 5. Conclusions

Our data demonstrate the feasibility and precision of CCTA for the follow-up of implanted Mg-BRS. Our results suggest good scaffold patency at 6 months, along with minimal lumen loss. Moreover, our data highlight the safety of Mg-BRS in the elective and ACS settings, with a slightly increased risk of MACE in the latter. In our cohort, one definite Mg-BRS thrombosis occurred despite optimal medical treatment, a rarely described event. Following an adequate selection of patients and implantation technique, Mg-BRS seems to be safe and efficacious in the long-term, with a word of caution for unstable patients. Further studies and improvements in CCTA imaging and Mg-BRS design might resolve some of these issues.

## Figures and Tables

**Figure 1 biomedicines-11-01150-f001:**
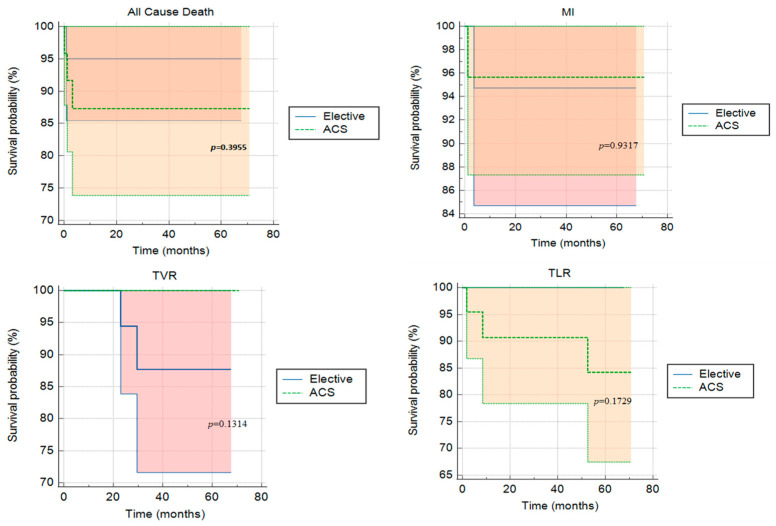
Kaplan–Meier survival curves between the elective and acute coronary syndrome (ACS) groups. MI = Myocardial Infarction; TVR = Target Vessel Revascularization; TLR = Target Lesion Revascularization.

**Figure 2 biomedicines-11-01150-f002:**
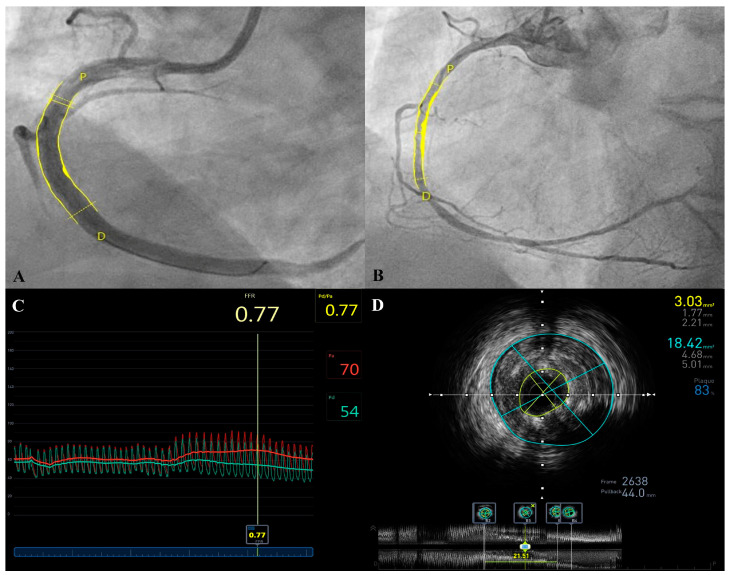
In-stent restenosis post-implantation of one DREAMS-2G stent in a right coronary artery lesion. MLD by QCA post-stent implantation was found to be 3.13 mm; reference 3.93 mm (Panel (**A**)). In-stent restenosis was found 8 months post-implantation with MLD by QCA of 1.2 mm (Panel (**B**)). Pathological functional assessment by FFR (0.77) was found upon revascularization (Panel (**C**)). IVUS pullback showed in-stent restenosis and an MLA of 3.03 mm^2^ secondary to a collapse of stent struts more than intrastent neointimal proliferation, the plaque burden was 83% (Panel (**D**)). FFR = Fractional Flow Reserve; IVUS = Intravascular Ultrasound; MLA = Minimal Lumen Area; MLD = Minimal Lumen Diameter; QCA = Quantitative Coronary Angiography.

**Figure 3 biomedicines-11-01150-f003:**
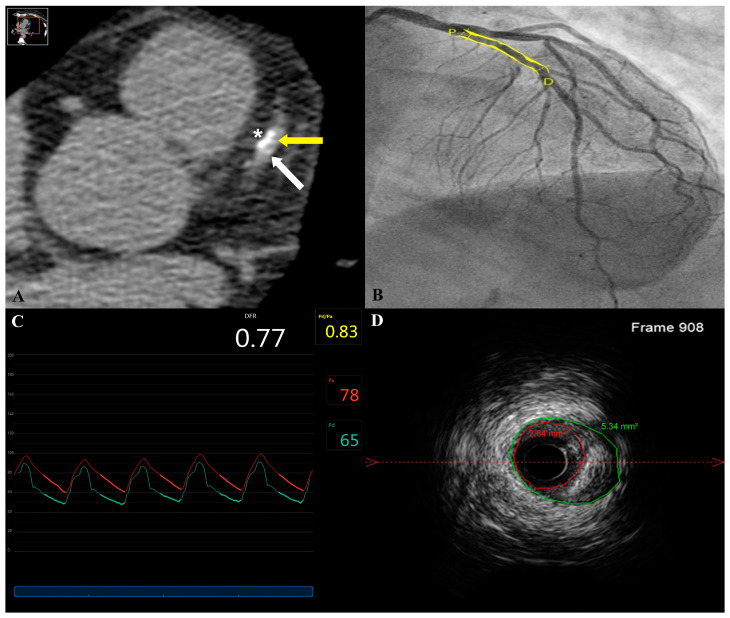
In-stent restenosis of a DREAMS-2G stent implanted in a proximal left anterior descending artery. In-stent restenosis by Coronary Computed Tomography Angiography 6-months post-stent implantation was suspected with a minimal in-stent lumen area (MLA) of 2.3 mm^2^. Panel (**A**) shows one slice of the CT without contrast, on which the proximal tantarum of the stent is visible (white arrow), the lumen of the artery (yellow arrow) is free from any metallic stent struts with blooming effect, while some parietal calcifications are visible (*). In-stent restenosis was confirmed by a coronary angiogram 8 months post-implantation with a minimal lumen diameter of 1.41 mm by quantitative coronary angiography (Panel (**B**)) and a pathological fractional flow reserve (0.77) (Panel (**C**)). An intravascular ultrasound (IVUS) pullback demonstrated in-stent restenosis with an MLA of 2.64 mm^2^ (Panel (**D**)).

**Figure 4 biomedicines-11-01150-f004:**
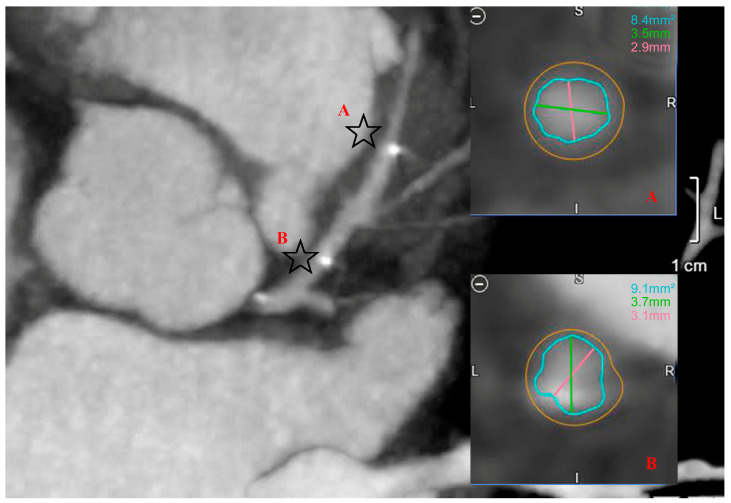
Cross sectional cut of a proximal left anterior descending artery and in-stent minimal lumen area. The two radiopaque spots (stars) are the tantarum markers at the proximal and distal edges of the stent with no discernible struts. Points A and B refer to cross sections at the respective levels.

**Figure 5 biomedicines-11-01150-f005:**
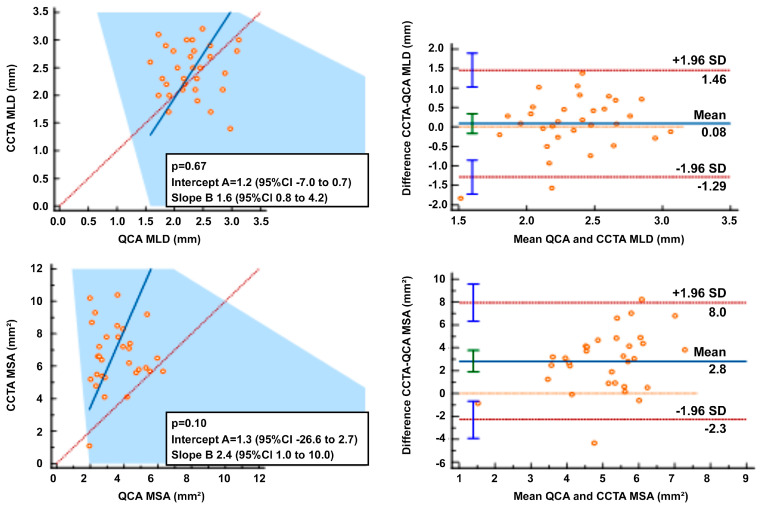
Quantitative Coronary Angiography (QCA) and Coronary Computed Tomography Angiography (CCTA) correlation and agreement. Passing-Bablok regression analyses on the left and Bland-Altman analyses on the right.

**Table 1 biomedicines-11-01150-t001:** Summary and comparison between currently available BRS.

Product	Manufacturer	Polymer	Drug	Strut Thickness (µm)	Resorption Time (Months)	Availability
Absorb	Abbot	PLLA	EES	150	48	Discontinued
Magmaris (DREAMS-2G)	Biotronik	Mg	SES	150	12	Available
DREAMS-3G	Biotronik	BIOmag	SES	120	12	Under study
Fantom	REVA Medical	Tyrosine polycarbonate	SES	125	36	Available
Fortitude	Amaranth	PLLA	SES	150	10	Available
Aptitude	Amaranth	PLLA	SES	115	10	Under study
Magnitude	Amaranth	PLLA	SES	98	36	Under study
MeRes	Meril	PLLA	EES	100	24	Under study
DESolve Cx	ELIXIR	PLLA	NES	120	24	Available
Firesorb	MircoPort	PLLA	SES	125	N/A	Under study
Mirage	Manli	PLLA	SES	150	14	Discontinued
Xinsorb	Biotech	PLLA	SES	160	24–36	Under study
NeoVas	Lepu	PLLA	SES	170	24–36	Available
Bioheart	Bioheart	PLLA	Rapamycin	125	6	Under study
Art	Terumo	PDLLA	None	170	6	Available
Renuvia	Boston Scientific	PLLA	EES	120	12–24	Discontinued
Unity	Amg	Mg/PLLA	SES	160	12	Under study

BRS: Bioresorbable scaffold; PLAA: Poly(L)-lactic acid; PDLLA: Poly(D,L)-lactic acid; Mg: Magnesium; EES: Everolimus-eluting scaffold; NES: Novolimus-eluting scaffold; SES: Sirolimus-eluting scaffold.

**Table 2 biomedicines-11-01150-t002:** General patient and procedure characteristics.

	Elective	ACS	Total	*p*-Value
n	20 (45.4%)	24 (54.5%)	44	
Age, years	60.5 ± 6.0	52.8 ± 7.3	56.3 ± 7.7	**0.001**
Male	19 (95.0%)	17 (70.8%)	36 (81.8%)	**0.038**
Hypertension	13 (65.0%)	13 (54.2%)	27 (61.4%)	**NS**
Diabetes Mellitus	8 (40.0%)	8 (33.3%)	16 (36.4%)	**NS**
PVD	2 (10.0%)	1 (4.2%)	2 (4.5%)	**NS**
Dyslipidemia	14 (70.0%)	14 (58.3%)	29 (65.9%)	**NS**
Renal failure	2 (10.0%)	0 (0.0%)	2 (4.5%)	**NS**
Family History of CAD	4 (20.0%)	7 (29.2%)	11 (25.0%)	**NS**
Current Smoking	7 (35.0%)	17 (70.8%)	23 (52.3%)	**0.017**
IVUS	7 (35.0%)	2 (8.3%)	9 (20.5%)	**0.029**
Lesions	25 (48.1%)	27 (51.9%)	52	
Treated vessel				
LAD	16 (64.0%)	17 (63.0%)	33 (63.5%)	NS
LCx	4 (16.0%)	2 (7.4%)	6 (11.5%)	NS
OM	2 (8.0%)	0 (0.0%)	2 (3.8%)	NS
RCA	3 (12.0%)	7 (25.9%)	10 (19.2%)	NS
RI	0 (0.0%)	1 (3.7%)	1 (1.9%)	NS
ACC/AHA classification				
A	10 (40.0%)	12 (44.4%)	22 (42.3%)	NS
B1	12 (48.0%)	6 (22.2%)	18 (34.6%)	NS
B2	3 (12.0%)	9 (33.3%)	12 (23.1%)	NS
Calcifications	8 (32.0%)	11 (40.7%)	19 (36.5%)	NS
Procedure characteristics				
n	25	27	52	
Pre-dilation diameter (mm)	2.65 ± 0.42	2.44 ± 0.30	2.55 ± 0.38	0.048
Mg-BRS diameter (mm)	3.10 ± 0.20	3.20 ± 0.25	3.15 ± 0.23	NS
Mg-BRS length (mm)	20.0 ± 4.6	21.3 ± 3.8	20.7 ± 4.2	NS
Mg-BRS inflation pressure (atm)	14.7 ± 1.0	15.1 ± 1.6	14.9 ± 1.4	NS
Post-dilation balloon diameter (mm)	3.38 ± 0.27	3.42 ± 0.29	3.41 ± 0.28	NS
Post-dilation inflation pressure (atm)	18.5 ± 2.6	18. ± 2.1	18.5 ± 2.3	NS
Post-procedure medications				
Clopidogrel	16 (80.0%)	7 (29.2%)	23 (52.3%)	0.001
Ticagrelor	4 (20.0%)	17 (70.8%)	21 (47.7%)	0.001
Events	4 (20.0%)	6 (25.0%)	10 (22.7%)	NS
Cardiac death	1 (5.0%)	3 (12.5%)	4 (9.1%)	NS
TV MI	1 (5.0%)	1 (4.2%)	2 (4.5%)	NS
TV-R	2 (10.0%)	0 (0.0%)	2 (4.5%)	NS
CD-TLR	0 (0.0%)	3 (12.5%)	3 (6.8%)	NS

PVD: peripheral vascular disease; IVUS: intravascular ultrasound; CPK: creatinine phosphokinase; CK-MB: creatinine kinase-MB; LAD: left anterior descending artery; LCx: left circumflex artery; OM: obtuse marginal artery; RI: ramus intermedius artery; Mg-BRS: Magnesium bioresorbable stent.

**Table 3 biomedicines-11-01150-t003:** Acute coronary syndrome general patients and procedure characteristics.

	UA	NSTEMI	STEMI	Total	*p* Value
n	5 (20.8%)	10 (41.7%)	9 (37.5%)	24	
Age, years	56.4 ± 2.9	51.1 ± 7.8	52.8 ± 2.4	52.3 ± 1.5	NS
Male	4 (80.0%)	6 (60.0%)	7 (77.8%)	17 (70.8%)	NS
Hypertension	3 (60.0%)	5 (50.0%	5 (55.6%)	13 (54.2%)	NS
Diabetes Mellitus	1 (20.0%)	4 (40.0%)	3 (33.3%)	8 (33.3%)	NS
PVD	0 (0.0%)	0 (0.0%)	1 (11.1%)	1 (4.2%)	NS
Dyslipidemia	1 (20.0%)	9 (90.0%)	4 (44.4%)	14 (58.3%)	0.02
Renal failure	0 (0.0%)	0 (0.0%)	0 (0.0%)	0 (0.0%)	
Family history	1 (20.0%)	4 (40.0%)	2 (22.2%)	7 (29.2%)	NS
Smoking	3 (60.0%)	7 (70.0%)	7 (77.8%)	17 (70.8%)	NS
IVUS	0 (0.0%)	2 (20.0%)	0 (0.0%)	2 (8.3%)	NS
Lesions	5	12	11	28	
Treated vessel					
LAD	1 (20.0%)	9 (75.0%)	7 (63.6%)	17 (60.7%)	NS
LCx	1 (20.0%)	0 (0.0%)	1 (9.1%)	2 (7.1%)	NS
OM	0 (0.0%)	0 (0.0%)	0 (0.0%)	0 (0.0%)	
RCA	3 (60.0%)	3 (25.0%)	2 (18.3%)	8 (28.6%)	NS
RI	0 (0.0%)	0 (0.0%)	1 (9.1%)	1 (3.6%)	NS
ACC/AHA classification					
A	4 (80.0%)	8 (66.7%)	1 (9.1%)	13 (46.4%)	
B1	1 (20.0%)	2 (16.7%)	3 (27.3%)	6 (21.4%)	
B2	0 (20.0%)	2 (16.7%)	7 (63.6%)	9 (32.2%)	
Calcifications	1 (20.0%)	4 (33.3%)	6 (54.5%)	11 (39.3%)	NS
Procedure characteristics					
n	4	12	11	27	
Pre-dilation balloon diameter (mm)	2.62 ± 0.18	2.45 ± 0.27	2.39 ± 0.34	2.44 ± 0.30	NS
Mg-BVS diameter (mm)	3.37 ± 0.25	3.21 ± 0.26	3.13 ± 0.23	3.20 ± 0.25	NS
Mg-BVS length (mm)	21.2 ± 4.8	20.8 ± 4.2	21.8 ± 3.4	21.3 ± 3.8	NS
Mg-BVS inflation pressure (atm)	15.5 ± 1.0	14.9 ± 2.2	15.1 ± 1.0	15.1 ± 1.6	NS
Post-dilation balloon diameter (mm)	3.69 ± 0.24	3.42 ± 0.29	3.34 ± 0.26	3.41 ± 0.28	NS
Post-dilation inflation pressure (atm)	18.5 ± 1.9	18.5 ± 1.9	18.2 ± 2.7	18.5 ± 2.3	NS
Post-procedure medications					
Clopidogrel	5 (100.0%)	1 (10.0%)	1 (11.1%)	7 (29.2%)	0.00
Ticagrelor	0 (0.0%)	9 (90.0%)	8 (88.9%)	17 (70.8%)	0.00
Events	2 (40.0%)	2 (20.0%)	2 (22.2%)	6 (25.0%)	NS
Cardiac death	1 (20.0%)	0 (0.0%)	2 (22.2%)	3 (12.5%)	NS
TV MI	0 (0.0%)	0 (0.0%)	1 (11.1%)	1 (4.2%)	NS
TVR	0 (0.0%)	0 (0.0%)	0 (0.0%)	0 (0.0%)	NS
CD-TLR	1 (20.0%)	2 (20.0%)	0 (0.0%)	3 (12.5%)	NS

PVD: peripheral vascular disease; IVUS: intravascular ultrasound; CPK: creatinine phosphokinase; CK-MB: creatinine kinase-MB; LAD: left anterior descending artery; LCx: left circumflex artery; OM: obtuse marginal artery; RI: ramus intermedius artery; Mg-BRS: Magnesium bioresorbable stent.

**Table 4 biomedicines-11-01150-t004:** Quantitative coronary analysis.

	Elective	ACS	Total	*p* Value
n	24	27	51	
Pre-treatment				
Lesion diameter (mm)	1.0 ± 0.4	1.2 ± 0.4	1.1 ± 0.4	NS
Proximal vessel diameter (mm)	2.7 ± 0.4	2.8 ± 0.6	2.7 ± 0.5	NS
Distal vessel diameter (mm)	2.4 ± 0.5	2.4 ± 0.5	2.4 ± 0.5	NS
Mean Area (mm^2^)	4.7 ± 2.0	5.5 ± 1.6	5.2 ± 1.8	NS
Post stenting				
Minimal diameter (mm)	2.4 ± 0.4	2.4 ± 0.5	2.4 ± 0.5	NS
Proximal vessel diameter (mm)	3.0 ± 0.5	2.9 ± 0.6	2.9 ± 0.5	NS
Distal vessel diameter (mm)	2.7 ± 0.5	2.9 ± 1.5	2.8 ± 1.1	NS
Minimal surface area (MSA) (mm^2^)	4.0 ± 1.8	4.3 ± 2.2	4.2 ± 2.0	NS
Mean proximal to distal surface area (mm^2^)	6.6 ± 2.0	6.6 ± 2.3	6.6 ± 2.1	NS

ACS = Acute Coronary Syndrome.

**Table 5 biomedicines-11-01150-t005:** Coronary computed tomography angiography.

	Elective	ACS	Total	*p* Value
n	15	17	32	
Date difference (months)	7.1 ± 4.5	5.6 ± 2.5	6.4 ± 3.7	NS
Proximal vessel diameter (mm)	4.0 ± 0.3	3.8 ± 0.7	3.9 ± 0.5	NS
Distal vessel diameter (mm)	3.3 ± 0.6	3.1 ± 0.6	3.2 ± 0.6	NS
Proximal surface area (mm^2^)	9.4 ± 1.9	10.0 ± 3.7	9.7 ± 2.9	NS
Distal surface area (mm^2^)	7.4 ± 2.1	7.1 ± 2.9	7.2 ± 2.5	NS
In-stent minimal diameter (mm)	2.4 ± 0.4	2.4 ± 0.7	2.4 ± 0.6	NS
In-stent maximal diameter (mm)	4.1 ± 0.5	3.9 ± 0.9	4.0 ± 0.8	NS
Minimal in-stent surface area (MSA) (mm^2^)	6.1 ± 1.2	6.8 ± 2.6	6.5 ± 2.0	NS
Maximal in-stent surface area (mm^2^)	11.3 ± 2.5	10.9 ± 3.4	11.1 ± 3.0	NS
Mean proximal to distal surface area (mm^2^)	8.4 ± 1.6	8.5 ± 2.9	8.4 ± 2.3	NS

ACS = Acute Coronary Syndrome.

## Data Availability

Data available on request.

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
