# Peer review of "Bioresorbable Magnesium-Based Stent: Real-World Clinical Experience and Feasibility of Follow-Up by Coronary Computed Tomography: A New Window to Look at New Scaffolds"

_biomedicines, 2023, doi:10.3390/biomedicines11041150_

Round 1

Reviewer 1 Report

The authors compared diagonostic accuracy in BRS by CCTA. The gold stanadard would by Angiography QCA. The authors used large paragraphs to described the patients' characteristics and clinical profiles, lesion characteristics. The authors should mention how many patients underwent repeated angiography and revascularization, and compared CCTA with QCA, and compared dianostic accuracy of stent patency or ISR in BRS or DES

Author Response

We thank the reviewer for his/her helpful insight and comments.

We would like to first point out that we believe that the gold standard to assess ISR is intravascular imaging. We have added two cases to illustrate this point (see figures 2 and 3)

We have truncated the large paragraph describing the patients’ characteristics since they are also given in tables 2-3

 “Between January 2017 and July 2020, a total of 52 Mg-BRS were implanted to treat 50 de novo coronary lesions in 44 patients who underwent primary PCI (Table 1). Mean patients’ age was 56.3 ± 7.7 years and there were 36 males (81.8%). Twenty-four (54.5%) patients presented with ACS among which 9 patients (20.9%) presented with ST-elevation myocardial infarction (STEMI). Intravascular ultrasound (IVUS) was used in 9 patients (20.5%): Patients presenting with ACS tended to be younger (p<0.05), males (p=0.05) and smokers (p<0.05).  A majority of left anterior descending (LAD) arteries (63.5%) were treated. One third of lesions had mild calcifications and a majority were B1-B2 according to the ACC/AHA classification. Mean implanted stent diameter size was 3.15±0.23] mm inflated at a mean pressure of 14.9±1.36 atm. Post-dilation was performed using a non-compliant balloon in all but three STEMI lesions (93.8%) at the discretion of the operator. Mean post-dilation balloon diameter was 3.4 ± 0.3 mm inflated at a mean pressure of 18.5 ± 2.4 atm. This reflected the adherence of the operators to the predefined Mg-BVS implantation technique recommendations [27,28].”

Repeated angiography and revascularization data can be found in section 3.2:

“Definite stent thrombosis by angiography occurred in 1 patient (4.2%) on day 37 post implantation. This patient who presented initially with an ACS was still on dual antiplatelet therapy. Three CD-TLR were reported in 3 patients (12.5%) of the ACS arm at 2, 8 and 52 months respectively. IVUS images showed in-stent neointimal hyperplasia treated by 1 DES implantation and severe plaque burden under the struts of the stent. The first 2 lesions were treated surgically whereas the latter showed angiographic as well as functional in-stent restenosis and was treated by DES implantation.”

Comparison between CCTA and QCA can be found in section 3.2:

“In 32 scaffolds, the mean difference between QCA and CCTA in MLD and MSA were -0.08 (limits of agreement (LOA) -1.29 to 1.46) and -2.84 (LOA -2.3 to 8.0) mm respectively.  Median minimal in-stent diameter by CCTA was 2.4 [2.1 – 2.8] mm while post PCI QCA minimal lumen diameter was 2.4 ± 0.5 mm, a non-significant statistical difference. Overall, there was no agreement between QCA and CCTA on a Passing-Bablok regression analysis”

This was also discussed in the discussion part: “In-scaffold luminal dimensions evaluation has already been compared by QCA and CCTA showing similar diagnostic accuracy[36]. Our QCA and CCTA measurements were performed on average 6 months apart and this might explain the lack of agreement reported. However, we confirm the feasibility to measure BRS luminal areas by CCTA and demonstrate accurate estimation of a loss of in-stent minimal diameter”

Unfortunately comparing the diagnostic accuracy of stent patency and ISR between BRS and DES is beyond the scope of this study as we did not randomize/match patients with DES to compare CCTA efficacy between the 2 groups (DES would leave a blooming effect as mentioned in the text hence preventing accurate measurements)

Reviewer 2 Report

The submitted manuscript is a retrospective, single-center study that evaluated 44 patients treated with Mg-BRS (bioresorbable magnesium based stents) between 2017 and 2020. During a median follow-up of 48 months, 10 events occurred, including 4 deaths. Bioresorbable magnesium based stents appeared to be safe and effective at long-term follow-up in both elective and acute settings according to the scaffold patency at 6 months. Overall, the study confirms the safety profile of Mg-BRS and the feasibility of follow-up with CCTA.

 Comments to be considered:

1. The manuscript presentation can be improved. Please, carefully check the manuscript for grammatical errors and typos.

2. Please, define all abbreviations since their first mention.

3. The quality of the tables and pictures is satisfactory. Yet, the authors may consider adding an additional figure showing the main characteristics of bioresorbable magnesium-based stents compared with other platforms.

4. The "blooming effect" is mentioned a few times in the text. The authors should explain this phenomenon in the text.

5. The text is sometimes challenging to understand for general cardiologists. The authors should consider adding a paragraph to clarify the practical implications of the study (e.g., on medical therapy).

6. The characteristics of coronary plaques (soft/calcific, bifurcations, long/short lesions) could influence the choice of stent and implantation techniques. The authors should explain whether these characteristics influenced the choice of stents.

7. Ischemic risk is considered high in patients undergoing PCI with bioresorbable stents. Please discuss the value of long-term dual antithrombotic therapy in this specific population making reference to recent articles on the topic: J Cardiovasc Pharmacol. 2020;76(2):173-180 (PMID: 32569017).

Author Response

We would like to thank the reviewer on the constructive and important points raised.

  • The text has been reviewed as suggested for English proofing and errors and typos were addressed throughout.
  • Abbreviations were reviewed and defined as noted.
  • A table comparing the different BRS available has been added as per your request
  • A small explanation of the blooming effect has been added in the introduction:

“blooming effect” defined as stent struts beam hardening artifacts resulting in artificial luminal narrowing and decreased intraluminal attenuation during non-invasive imaging with coronary computed tomography angiography”

  • And 6) Thank you for your valuable comment. We have added a paragraph addressing and summarizing both points:

“Our study highlights the safety of BRS implantation in a real world simple, soft and mildly calcified de novo lesion as recommended in BRS studies allowing for a transient radial force securing acute PCI results and resorbing with time. Furthermore, our data underlines the feasibility of CCTA follow-up of BRS, a non-invasive imaging tool, hence reducing the need for invasive coronary angiography for in-stent restenosis detection.”

DAPT duration discussion has been added and can be found in the discussion part: 

“Whereas dual antiplatelet therapy (DAPT) is recommended for at least 6 and 12 months post DES or BMS implantation in the elective and ACS setting, DAPT post BRS implantation remains currently recommended ≥ 12 months independently of implantation setting due to a higher rate of late stent thrombosis[49]. With an increasing number of reports lately towards shorter DAPT duration, data concerning BRS remain scarce and unconclusive, and a possible extension of treatment beyond 12 months could be reasonable extending from larger DES trials[50].”